# Divergent estimates of Miocene to Pleistocene upper ocean temperatures in the South Atlantic Ocean from alkenone and coccolith clumped isotope proxies

Heather M. Stoll[1], Clara Bolton[2], Madalina Jaggi[1,], Alfredo Martinez-Garcia [3], Stefano M. Bernasconi[1]

[1]Geological Institute, Department of Earth and Planetary Sciences, ETH Zurich, Zurich, 8092, Switzerland
[2]Aix Marseille Univ, CNRS, IRD, INRAE, CEREGE, Aix en Provence, France
[3] Max Planck Institute fur Chemie, Mainz, Germany

*Correspondence to*: Heather M. Stoll (heather.stoll@eaps.ethz.ch)

**Abstract.** Estimates of surface ocean temperatures in the past are essential for evaluating the sensitivity of Earth's surface temperature to higher atmospheric $CO_2$ concentrations such as those characterizing the Miocene. However, in the higher latitude regions, many proxy-based temperature estimates suggest extreme warmth, which imply much lower latitudinal temperature gradients than can be simulated by most coupled general circulation climate models under enhanced greenhouse gas forcing. This discrepancy implies either systematic biases in temperature proxy interpretation or the absence of key feedback processes in models. Here, we use a new approach to estimate high southern latitude surface ocean temperatures using clumped isotope thermometry in coccoliths - calcite plates precipitated in the surface ocean by the calcifying phytoplankton group coccolithophores. We present new determinations of the clumped isotope ratio in well-preserved coccoliths spanning the last 15 million years, extracted and purified from a sediment core located just south of the modern subtropical front (Ocean Drilling Program Site 1088, 41°S). Coccolith clumped isotopes reveal a 10°C decline in temperatures at this location over the last 15 million years, and over the last 11 Ma of overlapping records the magnitude of cooling is similar to that estimated from the degree of undersaturation of alkenone biomarkers. However, the temperatures derived from coccolith clumped isotopes are 8-12°C cooler than those estimated from alkenones, even though both are biosynthesised by the same organisms and therefore must reflect an identical production depth and season. This implies that some of the model-proxy mismatch may be due to unresolved issues in proxy interpretation. We propose that at this site, calibration biases lead to alkenone sea surface temperature estimates up to 5°C too warm, whereas coccoliths reflect temperatures at the production depth which is several degrees cooler than the sea surface. The influence of secondary diagenetic carbonate precipitation at the seafloor is constrained to contribute a cold bias of 2°C or less on the clumped isotope temperature for most samples.

## 1 Introduction

Under future greenhouse-induced warming scenarios, coupled general circulation models (GCM) project a greater warming of the high latitudes compared to lower latitudes (Masson-Delmotte et al., 2021). Yet, proxy records of past 'greenhouse' climates of the Miocene show an even greater high latitude amplification of warming, so that the weak latitudinal temperature gradients indicated by proxies are not well simulated by GCM using elevated $CO_2$ and past geography (Hossain et al., 2023) (Burls et al., 2021). Because temperature gradients on land and in the ocean significantly affect atmospheric circulation patterns including rainfall distribution (Burls and Fedorov, 2017), and the polar amplification is critical for predicting the stability of high latitude ice sheets (Gasson et al., 2013), this is a fundamental shortcoming with significant implications for predictions of future climate.

It is possible that climate models are missing key processes contributing to polar amplification: cloud physics parameterizations (Zhu et al., 2022; Zhu et al., 2019), ocean thermohaline circulation and heat transport (Hossain et al., 2020), and changes in sea ice parameterization all affect modeled temperatures and meridional temperature gradients. However, it is also feasible that models cannot reproduce the apparent features of these climates because the proxies are being systematically interpreted or calibrated incorrectly. Most estimates of the absolute surface ocean temperatures for past greenhouse periods derive from

analysis of changes in the structure or relative abundance of different classes of molecular fossils. These biomarkers, such as the unsaturation ratio of 37-chain alkenones $U_{37}^{k\prime}$ or the cyclizations in isoprenoid GDGTs (TEX$_{86}$) show statistically significant correlations with modern sea surface temperature (SST) in water column and core top studies and vary with temperature when source organisms are grown in culture experiments. Yet, the significance of biosynthesis processes in GDGT distributions and TEX$_{86}$ temperatures (Hurley et al., 2016; Polik et al., 2018), and the potential influence of selective degradation of more unsaturated alkenones and their effect on temperatures(Rontani et al., 2013; Kim et al., 2009b; Ausín et al., 2022) continue to be discussed.

To further evaluate the fidelity of higher latitude surface ocean temperature estimates, here we present data from a third surface ocean temperature proxy based on the frequency of dual substitution of the heavy isotopes of carbon and oxygen in carbonates, known as clumped isotope thermometry. Thermodynamic properties predict an increasing frequency of "clumped" heavy isotopes with lower temperature. In carbonates, the clumped isotope ratios are increasingly used for thermometry, and calibrations for abiogenically-precipitated carbonate (Anderson et al., 2021; Fiebig et al., 2021; Swart et al., 2021) as well as marine foraminifera (Meinicke et al., 2020; Peral et al., 2018; Daëron and Gray, 2023) and coccolithophorids (Clark et al., 2025; Clark et al., 2024a; Mejía et al., 2023) are now established.

We apply the clumped isotope thermometer to estimate the temperature during the production of coccoliths, microscopic plates of calcite produced by coccolithophorid algae in the ocean's photic zone. We focus on reticulofenestrid coccoliths produced by coccolithophores belonging to the Noelaerhabdaceae family, which includes the *Gephyrocapsa* species (including *G. huxleyi*, formerly *Emiliania huxleyi*) that dominate modern coccolithophore assemblages. The Noelaerhabdaceae family is also thought to be responsible for the production of alkenone biomarkers throughout the Neogene (Marlowe et al., 1984; Volkman et al., 1995; Volkman et al., 1980; Plancq et al., 2012). Thus, analysis of clumped isotopes in reticulofenestrid coccoliths allows us to compare the temperatures estimated from the organic biomarker and mineral phases produced by the same organism. This eliminates the complication of comparing temperature indicators from distinct organisms which may have different depth habitats within the photic zone or different seasonality of production. Here, we present new clumped isotope data on coccoliths isolated from two sites in the Subantarctic Zone of the South Atlantic Ocean spanning 15 to 0.8 Ma. In this region, alkenone temperature records indicate a substantial cooling (5-7°C) over the last 11 million years (Herbert et al., 2016).

## 2. Setting, Sediments and Analytical Methods

### 2.1 Sites, Sediments and Age Model

In this study, we use sediments from Ocean Drilling Program (ODP) Leg 177 in the southeastern Atlantic Ocean (Fig. 1). For the mid-Miocene to earliest Pleistocene, our samples are from ODP Site 1088 (41.13°S, 13.6°E; water depth of 2082m), well above the modern lysocline in this region and bathed today by North Atlantic Deep Water. Because Mid to Late Pleistocene sediments are depleted in Site 1088, we also analysed several samples from ODP Site 1090 (42.9°S, 8.8°E; water depth 3700 m), which is today bathed by Antarctic Bottom Water, in the Pliocene to late Pleistocene. Sediments at both sites consist of nannofossil ooze with variable amounts of foraminifera (Gersonde et al., 1999). Changes in Paleolatitude at Site 1088 have been modest since the mid-Miocene, with the site moving about 2.0° northward in the last 15 Ma (Herbert et al., 2016).

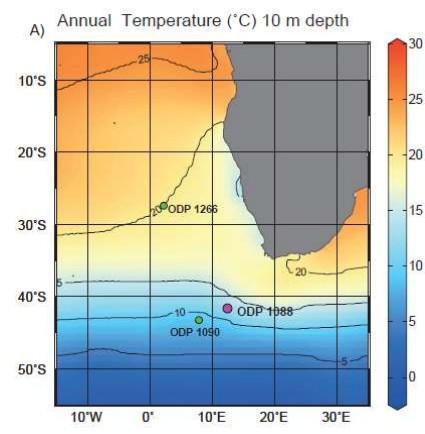

**Figure 1: Setting of Site 1088 and Site 1090 showing mean annual temperature at 10 m depth (color scale and contours). Also shown is Site 1266. Data from World Ocean Atlas (Locarnini et al., 2013). Map from (Tanner et al., 2020).**

Site 1088 is located in the modern Subantarctic Zone which is defined as lying between the Subtropical Front (STF) and the Subarctic Front (SAF). Near site 1088 (40.5°S), modern SST varies seasonally between 13 and 15°C while temperature at 100

m depth varies from 12 to 13°C (Locarnini et al., 2013). Near the location of Site 1090 (42.5°S), modern SST ranges from 9 to 11°C while temperature at 100 m depth varies from 8.5 to 10°C.

For Site 1088, we employ the previously published age model (Herbert et al., 2016) which is based on calcareous nannofossil biostratigraphy (Marino and Flores, 2002) as well as isotope stratigraphy (Billups et al., 2002). For Site 1090, we employ a published age model (Venz and Hodell, 2002), which is identical to that employed for published alkenone temperatures at that site (Martínez-Garcia et al., 2009).

## 2.2 Sample preparation and analysis

For sediments from Site 1088, narrow coccolith size-fractions were isolated from the <20 µm fraction of nine bulk sediment samples using repeated decanting and microfiltration protocols described in detail in reference (Bolton et al., 2012; Bolton and Stoll, 2013). In this study, we focus on the 2.5 to 4 µm fraction, which is dominated (>78% by mass) by reticulofenestrid coccoliths and for which stable isotope results, assemblage composition, and preservation were already reported (Bolton and Stoll, 2013). This analysis verifies that there is no significant non-coccolith biogenic or abiogenic carbonate in this size fraction. For comparison, for one sample around 14 Ma, we additionally analyzed the larger coccolith size fraction (6-9 µm) which is dominated by *Cyclicargolithus floridanus* ( a larger reticulofenestrid) and *Coccolithus pelagicus*. Detrital carbonate fragments are rare in the examined samples from Site 1088. For sediments from ODP Site 1090, we microfiltered five bulk sediment samples to obtain a <5 µm or <8 µm fraction strongly dominated by reticulofenestrid coccoliths in all but one sample, which is dominated by *Coccolithus pelagicus*. Assemblage composition for all samples is detailed in Supplementary Tables 1 and 2 and illustrated in light microscope images in Figures S1 and S2.

To reduce potential interferences by organic phases during measurement, sediments were oxidized with 10% $H_2O_2$ buffered solution with $NH_3$ to pH=8-9 and the oxidant removed. Where the abundance of material in the size fraction allowed, two oxidation times were compared, a 5-hour long oxidation and an 8-14 hour long oxidation, and where material was limiting the 5-hour long oxidation was performed. Subsequently sediments were thoroughly rinsed at least three rinses with ultrapure (Milli-Q) water and dried overnight at 60°C. Following convention for carbonates, we report the relative clumped isotope ratio as the $\Delta_{47}$. The $\Delta_{47}$ of the prepared sediment samples were measured on a Kiel IV-MAT 253Plus system (ThermoFisher Scientific, Bremen, Germany) following the LIDI protocol (Hu et al., 2014; Müller et al., 2017), using sample and carbonate standards (ETH-1, 2, 3, and IAEA C2) aliquots yielding 100-110 µg of $CaCO_3$ per replicate. The Kiel IV device purifies liberated $CO_2$ through a PorapakQ held at -40°C. Pressure dependent backgrounds for correction of non-linearity effects were measured before each batch of 44 samples/standards was started (Bernasconi et al., 2013; Meckler et al., 2014). Data reduction and baseline corrections were carried out using the Easotope software (John and Bowen, 2016), and raw carbon and oxygen values were converted to VPDB using the $^{17}O$ correction parameters as recommended by previous studies (Daëron et al., 2016). The $\Delta_{47}$ values are reported in the Intercarb carbon dioxide equilibrium scale (I-CDES) by normalization to the ETH-1, ETH -2 and ETH -3 standards as described previously (Bernasconi et al., 2021). Each analytical run consists of 10 aliquots of ETH-3, 5 aliquots each of ETH-1 and ETH-2 organized in three blocks: one at the beginning, one in the middle and one at the end of the autosampler carousel, two aliquots of IAEA-C2 and not more than 3 aliquots each of unknown samples for a total of 22 aliquots. For each sample from Site 1088 a minimum of 14 replicates and for 1090 a minimum of 10 replicates distributed in 4 to 5 analytical runs was measured. For standardization, a moving window of 12 standards measured before and 12 standards measured after each sample (spanning two to three analytical runs) was used. The IAEA-C2 international carbonate standard used to monitor long-term accuracy and reproducibility during the measurement period yielded a $\Delta_{47} = 0.6410 \pm 0.024$ ‰ (1σ, n = 229) in very good agreement with the accepted value of 0.6409 ‰ ± 0.003 ‰ (Bernasconi et al., 2021). Analytical errors are reported at the 95% confidence interval (Fernandez et al., 2017).

Temperatures were calculated from measured $\Delta_{47}$ using the cultured coccolith calibration (Clark et al., 2024a). This calibration has been validated by comparison of sediment trap coccolith values with temperature at the depth of maximum

coccolithophore abundance in the photic zone(Clark et al., 2025; Clark et al., in review). The $\Delta_{47}$ temperature reported here therefore represents the temperature at the depth and season of coccolithophore production. We compare the $\Delta_{47}$ temperatures with published temperatures calculated from alkenone unsaturation ratios. Alkenone temperatures for Site 1088 (Herbert et al., 2016) are calculated with the $U^{k\prime}_{37}$ index based on the C37:2 and C37:3 alkenone abundances; no C37:4 abundances are reported for these samples. This published record employed the calibration based on regression of sea surface temperatures to globally distributed core top $U^{k\prime}_{37}$ (Müller et al., 1998). Alkenone temperatures for Site 1090 (Martínez-Garcia et al., 2011) employ the $U^{k}_{37}$ index which is based on the C37:2, C37:3, and C37:4 abundances. This latter index is proposed to be better suited for colder temperature settings(Ho et al., 2012) as long as contributions from non-marine haptophytes can be ruled out(Kaiser et al., 2019). The temperature calculation at 1090 employs a calibration based on culture of a strain of *Gephyrocapsa (Emiliania) huxleyi*(Prahl et al., 1988) which is not significantly different from the calibration obtained from regressions between SST and $U^{k}_{37}$ in core top sediments (Sikes et al., 1991; Ho et al., 2012)

## 3 Results

### 3.1 Coccolith $\Delta_{47}$ temperatures at ODP 1088 and 1090

At Site 1088, the $\Delta_{47}$ temperatures from the reticulofenestrid size-fraction range from a Pleistocene minimum of 4.6°C (± 1.8) to a mid-Miocene maximum of 14.6°C (± 1.9), defining a 10°C cooling trend since the mid Miocene (Fig. 2a). A local minimum of 7.5°C at 5.6 Ma marks the culmination of the Late Miocene Global Cooling. In Site 1088, coccolith replicates treated with a longer oxidation time yield temperatures within error of the 5h reaction time applied to the full sample set, with samples at 3 and 4 Ma showing near-identical means, while the youngest sample (0.8 Ma) has a 2.5°C higher mean temperature with longer oxidation (Fig. 2a). The sample from the 6-9 μm fraction yields an $\Delta_{47}$ temperature indistinguishable from that of the smaller reticulofenestrid sample when treated with the same oxidation protocol. When treated with the longer oxidation protocol, the estimated $\Delta_{47}$ temperature of the 6-9 μm fraction increases. Longer oxidation increases the selective dissolution experienced by the sample. In samples with heterogeneous species composition, more intense dissolution may lead to shifts in the species composition to more dissolution resistant taxa (Gibbs et al., 2004). If the more dissolution resistant species have an ecological niche favoring warmer parts of the seasonal cycle or even warmer conditions across interannual variability, then increased dissolution may lead to warmer measured $\Delta_{47}$ temperatures.

At Site 1090, the Plio-Pleistocene $\Delta_{47}$ temperatures from the reticulofenestrid size fraction average 7°C and range from 5 to 9 °C (Fig. 2b).Three samples contain minor detrital carbonate (Fig. 2b, Fig. S1) Wind transport of detrital minerals to this location via strong westerlies has been documented from Patagonia and southern South America as well as Southern Africa (Barkley et al., 2024). For the cold coccolith temperatures (5 °C), small amounts (<5%) of detrital carbonate would warm the measured clumped isotope temperatures by <1.5°C if the clumped isotope signature of detrital carbonates reflected temperatures of carbonate precipitation in earth surface environments at temperatures <35°C. However, if the source of the detrital carbonates has had clumped temperatures reset to much higher burial temperatures (>100°C) it could shift the measured temperatures warmer by several degrees in coccolith fractions containing >3% of such detrital carbonates. Since the origin and burial history of the detrital carbonates cannot be readily constrained for this setting, and the detrital carbonates cannot be effectively isolated from the coccoliths for analysis of their clumped temperature, it is not possible to predict if they appreciably impact the measured temperatures in the samples containing them. As a conservative approach, we do not make further interpretations from these samples.

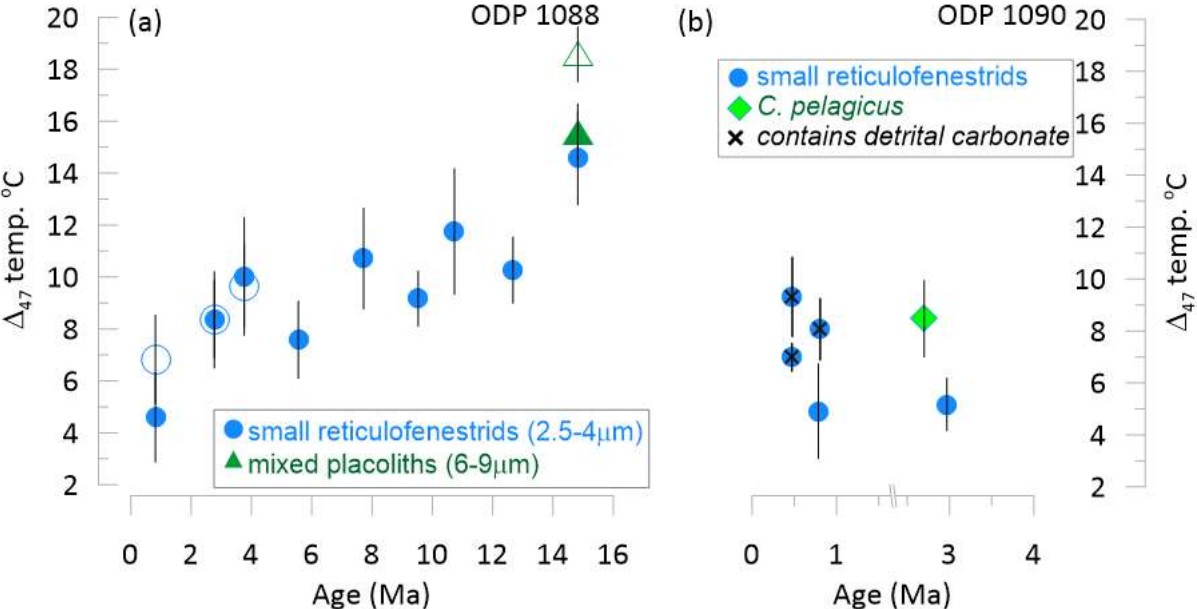

Figure 2: New clumped isotope temperatures from coccoliths, distinct symbols illustrate the coccolithophore family dominating the measured assemblage. (a) $\Delta_{47}$ temperature from Site 1088. Open symbols indicate longer oxidation, filled symbols indicate 5 hour oxidation. (b) $\Delta_{47}$ temperature from Site 1090; symbols with x contain 2 to 5% detrital carbonate.

### 3.2 Comparison of coccolith $\Delta_{47}$ temperatures, published alkenone temperatures, and water column temperatures

The $\Delta_{47}$ coccolith temperatures are compared with temperatures derived from alkenones in Fig. 3. At ODP 1090, the $\Delta_{47}$ temperatures of the reticulofenestrid-dominated coccolith fractions are within the range of $U^{k}_{37}$ SST (Fig. 3d). For the three samples with minor detrital carbonate, $\Delta_{47}$ temperatures are 1 to 4 °C warmer than $U^{k}_{37}$ SST. For the other two reticulofenestrid dominated samples the $\Delta_{47}$ temperatures are 2.3 to 5.4 °C cooler than $U^{k}_{37}$ SST. Compared with the alkenone temperatures averaged in an interval +/-5 ky or +/- 10 ky around the age of the $\Delta_{47}$ temperatures, the $\Delta_{47}$ temperature of the reticulofenestrid-dominated coccoliths average 2.7 and 2.4 °C cooler, respectively. Coccolith $\Delta_{47}$ temperatures from the last 3 Ma fall between estimated glacial summer faunal temperatures(Becquey and Gersonde, 2002) and modern water column temperatures during the spring-summer production period (Fig. 3c,d).

At Site 1088, over the last 11 Ma for which alkenone SST estimates are available, reticulofenestrid coccolith $\Delta_{47}$ temperatures average 9°C cooler than the alkenone-derived SST estimates (Fig. 3b). Compared to Site 1090, the sample resolution of the Site 1088 $U^{k\prime}_{37}$ is lower so we estimate the SST over a longer time window of +/-0.2 Ma. When compared against either the stratigraphically closest alkenone SST or the alkenone SST averaged over +/-0.2 Ma from the $\Delta_{47}$ temperatures, the temperature offset between coccolith $\Delta_{47}$ temperatures and the alkenone SST at Site 1088 over the last 11 Ma is relatively constant (Fig. 3e). The youngest coccolith $\Delta_{47}$ sample with long oxidation, which falls in a glacial period, features a temperature about 5°C cooler than the modern lower euphotic zone temperatures (7°C cooler for the short oxidation replicate). The short oxidation replicate gives a temperature similar to faunal summer surface temperature estimates at Site 1090 (Fig. 3d(Becquey and Gersonde, 2002)). Over the last 12 Ma, with the exception of the minimum at ~6 Ma culminating the Late Miocene cooling, the alkenone SST is always comparable to or warmer than modern warm season SST at the site (Fig. a-b). Over the last 15 Ma, the $\Delta_{47}$ temperatures are all comparable to or colder than the modern 150 m depth warm season temperature, except for the 15 Ma sample which is comparable to modern warm season surface temperatures (Fig. 3a-b).

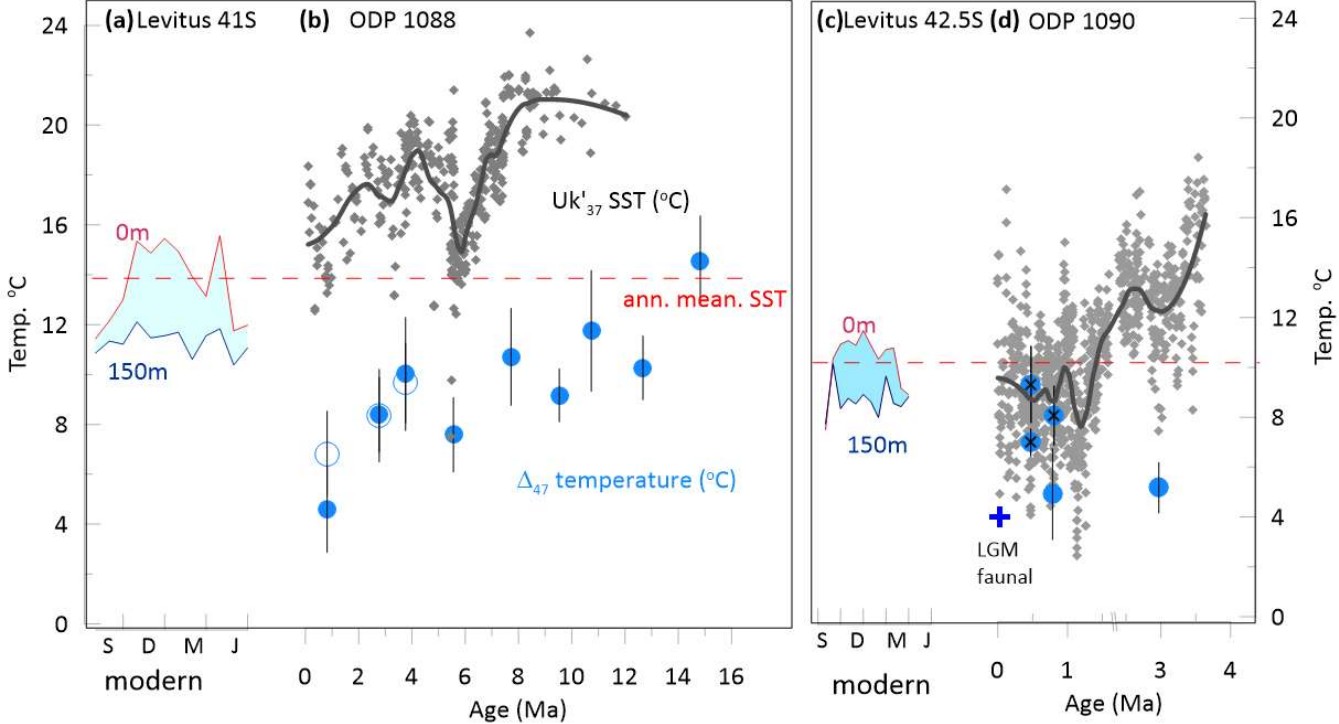

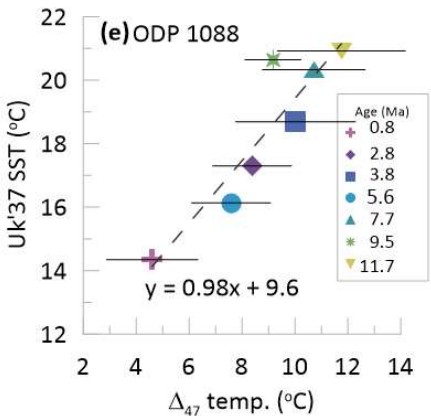

**Figure 3. (a) (c)** Modern seasonal temperature evolution near sites 1088 (a) and 1090 (c) beginning in southern hemisphere spring (S=September, D=December, =March, J= June) for water depths of 0m and 150m(Levitus et al., 2010), with indication of modern annual mean SST **(b) (d)** comparison of published $U^{k\prime}_{37}$ for Site 1088 ((Herbert et al., 2016)) and $U^{k}_{37}$ for Site 1090(Martínez-Garcia et al., 2011) with the $\Delta_{47}$ coccolith temperature estimates of reticulofenestrid dominated fractions as illustrated in Fig. 2. In Site 1088 and 1090 a LOESS fit is illustrated for the $U^{k\prime}_{37}$ records. For Site 1090, the LGM faunal summer temperature estimate (Becquey and Gersonde, 2002) is shown. **(e)** For Site 1088, reticulofenestrid dominated $\Delta_{47}$ coccolith vs. the $U^{k\prime}_{37}$ SST estimate in the 0.4 m.y. age bin containing the $\Delta_{47}$ coccolith analysis, and the linear regression equation, which has $r^2$=0.84 and p-value <0.005. Symbols indicate the sample age and 68% error bars are shown on $\Delta_{47}$ temperatures.

## 4. Discussion

### 4.1 Factors potentially contributing to divergent alkenone and coccolith temperatures

#### 4.1.1 Coccolith and alkenone production in spring bloom domains

We present analyses of the coccoliths of the reticulofenestrid lineage which produces alkenones. During cell growth, coccoliths are continually produced in order to provide a covering of both daughter cells following cell division (Suchéras-Marx et al., 2022) while alkenones are continually synthesized by cells for carbon and energy storage (Tsuji et al., 2015). Cruise data suggest that modern reticulofenestrid coccolithophores (*Gephyrocapsa spp.* including *G. huxleyi*) are lower euphotic zone dwellers with maximum abundance at the depth of between 1 and 10% of surface irradiance (Poulton et al., 2017). In mid- and high-latitude regions characterized by deep winter mixing, peak coccolithophore production and export occurs in a spring

bloom (Haidar and Thierstein, 2001; Broerse et al., 2000; Ziveri et al., 2000). Monthly sampling of the water column at the location of the Bermuda Atlantic Time Series (31°50'N 64°10'W) shows that during the spring bloom, high numbers of alkenone producers are found throughout the upper 100 m of the water column, with significant cell numbers extending down to 150 m (Haidar and Thierstein, 2001).

Because coccoliths and alkenones are produced during growth of the same organism, the differences in calculated temperatures cannot reflect differences in production depth or seasonality of proxy carriers. However, the most commonly employed calibration practices differ for $\Delta_{47}$ and alkenone unsaturation. The calibration of $\Delta_{47}$ to temperature is made using calcification temperature based on experimentally grown cultures, and therefore reflects the temperature at which coccolithophorids grow and calcify in the photic zone, in this case during the bloom season. In contrast, the $U_{37}^{k\prime}$ (or $U_{37}^{k}$) ratio has been empirically calibrated to the mean annual temperature at the sea surface (0 m) regardless of the actual production depth (Müller et al., 1998; Tierney and Tingley, 2018), or in some regions calibrated to summer temperatures (Tierney and Tingley, 2018). This calibration approach assumes that vertical or temporal differences are similar at all sites used in the calibration and where the calibration is applied. Thus, there is the potential for an offset in the $\Delta_{47}$ and alkenone temperatures whenever coccolithophore production occurs deeper than the surface mixed layer, in waters with cooler temperatures.. At the location of Site 1088, the temperature difference between 100 m and the surface is <3°C even during the spring to early summer season expected to host the most significant coccolithophorid production. Therefore, at this setting in modern oceanographic conditions, the distinct temperature calibration targets may explain up to 3°C difference but are insufficient to explain the full 3°C difference between $\Delta_{47}$ and coccolith temperatures.

In some marine settings like the Baltic sea region of unusually low salinity (Kaiser et al., 2019) and Arctic settings (Wang et al., 2021), noncalcifying haptophytes also make significant contributions to sedimentary alkenones. However, in Late Miocene sediments at Site 1088, the distribution of C38 methyl and ethyl alkenones is consistent with a dominant contribution from Group III marine haptophytes which represent the calcifying reticulofenestra and no significant contribution from noncalcifying Group II or Group I alkenone producers (Tanner et al., 2020).

### 4.1.2 Potential for bias in $\Delta_{47}$ coccolith due to seafloor diagenetic overgrowth of coccoliths

At Site 1088, like most regions of the ocean, upper ocean temperatures are warmer than those at the seafloor (Locarnini et al., 2013). If the measured coccolith carbonate represented a mixture of primary biogenic calcite produced in the euphotic zone and secondary diagenetic carbonate precipitated in the colder waters on the seafloor, this could cause coccolith $\Delta_{47}$ temperatures to be significantly colder than those in the euphotic zone. However, in the higher latitudes such as Site 1088, because of the relatively modest temperature gradient between the seafloor and the euphotic zone (Locarnini et al., 2013), extreme degrees of diagenetic overgrowth would be required to significantly shift the temperature signal of the measured carbonate.

SEM images from Site 1088 indicate generally very good preservation. In samples from the Plio-Pleistocene, there is no evidence of secondary infilling or overgrowth of the primary placolith crystals, whereas in the older Miocene samples, there is minor infilling of central area structures in the reticulofenestrids ((Bolton and Stoll, 2013) Figs S6, S7). The most extreme cold temperatures recorded by coccolith $\Delta_{47}$ in our record occur in the Pleistocene, where diagenetic overgrowth is not evident. Considering the 0.8 Ma glacial period recorded by the youngest $\Delta_{47}$ sample, if the true SST value were 14°C as calculated from the alkenones, even if the coccoliths were produced 100 m deeper at 11°C, to produce calcite with a weighted average $\Delta_{47}$ temperature of 4.6°C would require that 70% of the measured carbonate was diagenetic precipitated at a seafloor temperature of 2°C and only 30% of the measured carbonate was pristine and precipitated at 11°C. This extent of diagenetic contribution is incompatible with SEM and light microscope images of coccoliths.

Furthermore, measured coccolith stable isotope and Sr/Ca ratios on the same reticulofenestrid 2.5 to 4 µm fraction rule out a major contribution from secondary carbonate. The coccoliths from the Plio-Pleistocene at Site 1088 preserve evidence of

persistent vital effects of 1.5-3 ‰ in $\delta^{18}O$ and $\delta^{13}C$ that were driven by $CO_2$ decline (Bolton and Stoll, 2013). The maintenance of significant isotopic heterogeneity among coccolith size fractions is incompatible with a large contribution of common secondary seafloor carbonate, which would homogenize the primary biogenic vital effects. The temperature offset between alkenones and coccolith $\Delta_{47}$ is of similar magnitude during the older period of very limited interspecific coccolith vital effects and the last 5 Ma period of significant vital effects (Fig. 3e).

Coccolith Sr/Ca is a particularly potent indicator of secondary diagenetic carbonate because abiogenic Sr partitioning is an order of magnitude lower than that of coccolithophore biomineralization (Tang et al., 2008). Sr/Ca ratios of $\approx 0.26$ mmol/mol are predicted for precipitation from modern seawater. Reticulofenestrid coccoliths produced in the modern spring bloom setting of the North Atlantic with average water column temperature of 19°C, sampled in sediment traps, feature Sr/Ca ratios of 2.5 mmol/mol. Culture studies show that the primary Sr/Ca ratios of reticulofenestrid coccoliths increases by 0.05 mmol/mol for

every 1 °C increase in calcification temperature (Stoll et al., 2002a; Müller et al., 2014; Stoll et al., 2002b). We thus estimate the expected primary Sr/Ca of reticulofenestrid coccoliths produced at the Site 1088 setting in the past, inferring that productivity was constant and comparable to the modern spring bloom regime at Bermuda, but adjusting for the temperature dependence of Sr partitioning in coccoliths with either the assumption of production temperature 3°C colder than reported alkenone temperatures, or the assumption of production temperature equivalent to measured with $\Delta_{47}$ coccolith. This

calculation of primary Sr/Ca assumes that no processes other than temperature changes caused temporal changes in the biogenic Sr incorporation. We subsequently calculate by mass balance, the Sr/Ca ratio of mixtures of this primary coccolith carbonate and secondary carbonate precipitated abiogenically at seafloor temperatures assuming seawater Sr/Ca has remained close to modern over the last 15 Ma as suggested by records of seawater chemistry(Lear et al., 2003) .

       In the case that coccolith production temperatures were 3°C cooler than SST estimated from alkenone unsaturation, the

measured Sr/Ca ratios of the Site 1088 reticulofenestrid fractions (Bolton and Stoll, 2013) are compatible with 0 to 20% of secondary diagenetic carbonate precipitation (Fig. 4a) for all except two samples (7.5 and 9.5 Ma) which are compatible with 30% and 25% secondary diagenetic carbonate, respectively (Fig. 4a). If the coccolith production temperatures are well reflected by the $\Delta_{47}$ coccolith, then the measured Sr/Ca ratios are compatible with 0 to 20% secondary diagenetic carbonate (Fig. 4b). For either 8°C or 11°C temperature gradient between the seafloor and production depth, modest (20%) contributions

of secondary diagenetic carbonate would cool $\Delta_{47}$ temperatures by only 1.7 to 2.6°C (Fig. 4c). Thus, secondary carbonate precipitation at colder seafloor temperatures is not sufficient to generate the observed 9°C average offset between $\Delta_{47}$ and the $U_{37}^{k'}$ temperatures.

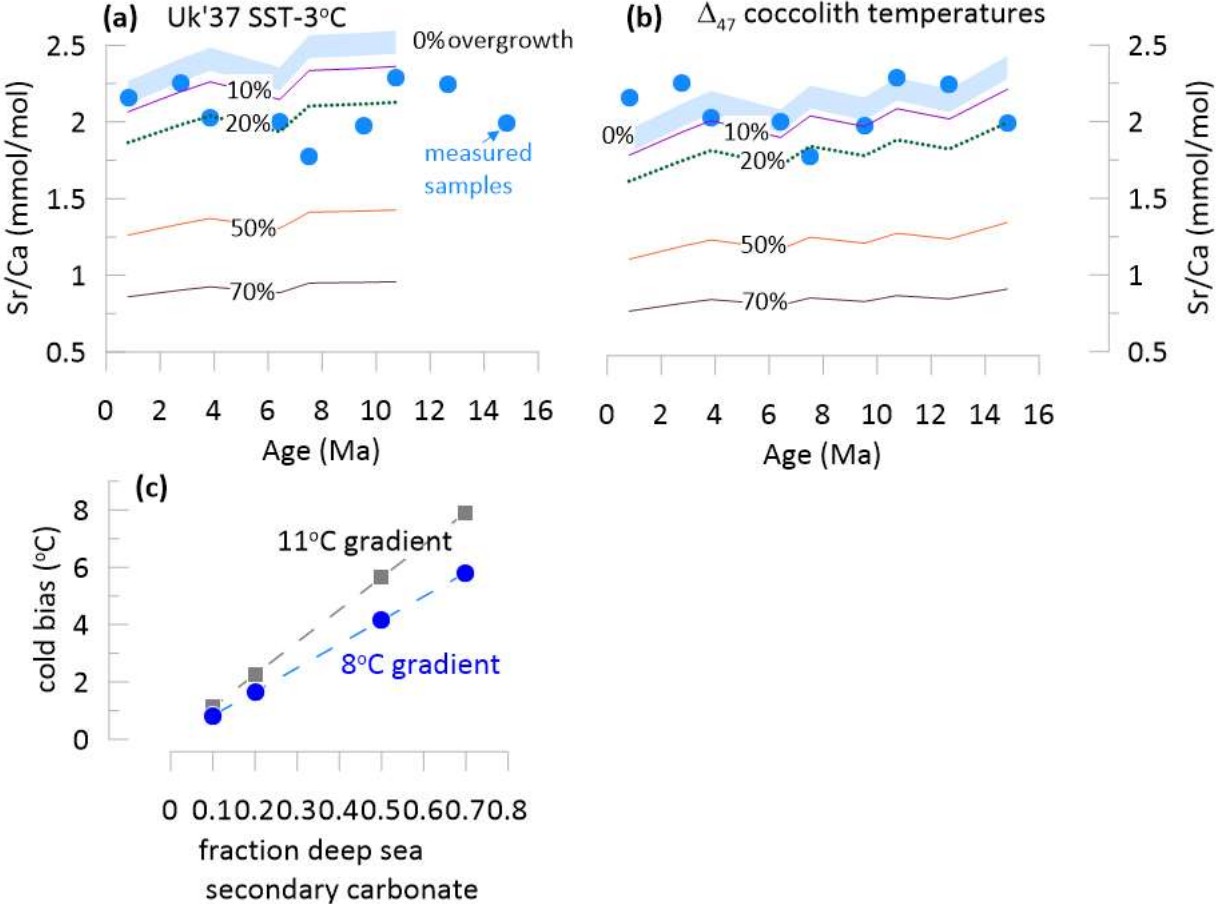

Figure 4. (a) (b) Measured Sr/Ca ratios in the 2.5-4 μm fraction from Site 1088 (Bolton and Stoll, 2013), compared to predicted range of primary Sr/Ca ratio of small Reticulofenestra with no overgrowth and with varying fractions of the measured sample corresponding to secondary diagenetic overgrowth. (a) indicates prediction of primary Sr/Ca for each sample using a coccolith production temperature 3°C cooler than the $U_{37}^{k\prime}$ SST. (b) indicates prediction of primary Sr/Ca for each sample using the measured $\Delta_{47}$ coccolith temperatures. Full details of the calculation are given in section 4.1.2. (c) Magnitude of cold bias induced by different fractions of calcite from secondary seafloor precipitation, assuming either a temperature difference between primary biogenic and secondary calcite equivalent to modern SST to deep ocean temperatures (11°C), or modern 150 m to deep ocean temperatures (8°C).

**4.1.3 Selective degradation of C37:3 alkenones in oxic bottom waters**

While sedimentary processes have the potential to introduce a cold bias to carbonate-based $\Delta_{47}$ (and $\delta^{18}O$) signals from coccoliths and other planktic carbonates, water column and sedimentary processes have the potential to introduce a warm bias to the biomarker alkenone indicator. Selective degradation of more unsaturated alkenones occurs by both autooxidation and aerobic bacterial degradation in oxic environments. This process alters the the $U_{37}^{k\prime}$ (and $U_{37}^{k}$ ) ratios and can lead to warm temperature biases of up to 5.9 °C (Rontani et al., 2013) (Kim et al., 2009b; Ausín et al., 2022). The influence of selective degradation in the water column and at the sediment-water interface would be accounted for in core top calibrations, and has been discussed as a contributing factor to the difference between suspended particulate matter and core top alkenone unsaturation index (Conte et al., 2006). If selective oxidation subsequently continued at significant rates over millions of years in the sediments after deposition, it would be expected to lead to greater divergence between the $U_{37}^{k\prime}$ and coccolith $\Delta_{47}$ in older sediments. This is not observed in our records from Site 1088. It is possible that selective degradation of alkenones in sediments occurs only early in the sedimentation process, when oxygen content in sediments is highest. While not leading to long term increases in the $U_{37}^{k\prime}$, if this process were continuing over the upper few meters of sediment, it may nonetheless contribute further selective degradation and increase in the $U_{37}^{k\prime}$ which is not encompassed in the core top calibration.

### 4.1.4 Differential transport of alkenones and coccoliths

The persistence of alkenones in sediments is attributable to the stabilization of the lipids on mineral surfaces, in particular on the large surface areas of clay and silt sized particles. The alkenone-carrying particles may be subject to resuspension and transport. For example, in the deepest part of the Cape Basin, where sediment drifts attain accumulations of >120m/my in the late Pleistocene at Site 1089 (4620 m(Gersonde et al., 1999)), bottom currents lead to a significant transport of alkenones produced in warmer, lower latitude surface waters to final deposition in the deep Cape Basin against the Agulhas Ridge. This transport is estimated to contribute to a 6°C warmer temperature in sedimentary core-top alkenones compared to the directly overlying SST (Sachs and Anderson, 2003; Kim et al., 2009a). However, in the shallower sites ODP 1088 and ODP 1090 (water depths 2082 and 3702 m respectively), sedimentation rates in the Pleistocene are significantly lower (over the last 1 Myr, ~30 m/my at Site 1090 and 12 m/myr at Site 1088(Gersonde et al., 1999)), suggesting less sediment focusing. Nonetheless, the potential for transport at these sites has not been evaluated. In this Southern Ocean setting, it has also not been assessed whether the large fraction of the alkenones reside in the very fine fraction sediment fraction or a larger silt fraction.

If the majority of the alkenones reside on particles in a similar grain size as the coccoliths (<15μm), then it might be expected that similarly shaped and sized particles would experience similar hydrodynamic transport. Unless individual coccoliths dissolve in transport, they may be (re) transported from the same source regions as alkenone-bearing clays and therefore expected to carry the same proxy signal. If on the other hand, alkenones are bound to larger particles that experience a different transport regime (i.e., a sortable silt regime rather than a cohesive transport regime), then it is possible that the transport of alkenones and coccoliths may be decoupled, leading to divergent surface ocean source regions for the two proxy phases buried in the same sediment core. Alkenone and coccolith sources may also be partially decoupled if the fractional loss during deep export from the photic zone were much greater for alkenones than coccoliths, elevating the significance of laterally transported alkenones compared to that of coccoliths. Nonetheless, given the similar long-term trends, if differential transport of coccoliths and alkenones was responsible for the offset in temperatures, it would require a sustained temperature difference in the source areas and require similar transport pathways over the last 15 Ma. This is unlikely given reorganizations of Southern Ocean deep and surface ocean circulation within that timeframe (Evangelinos et al., 2024).

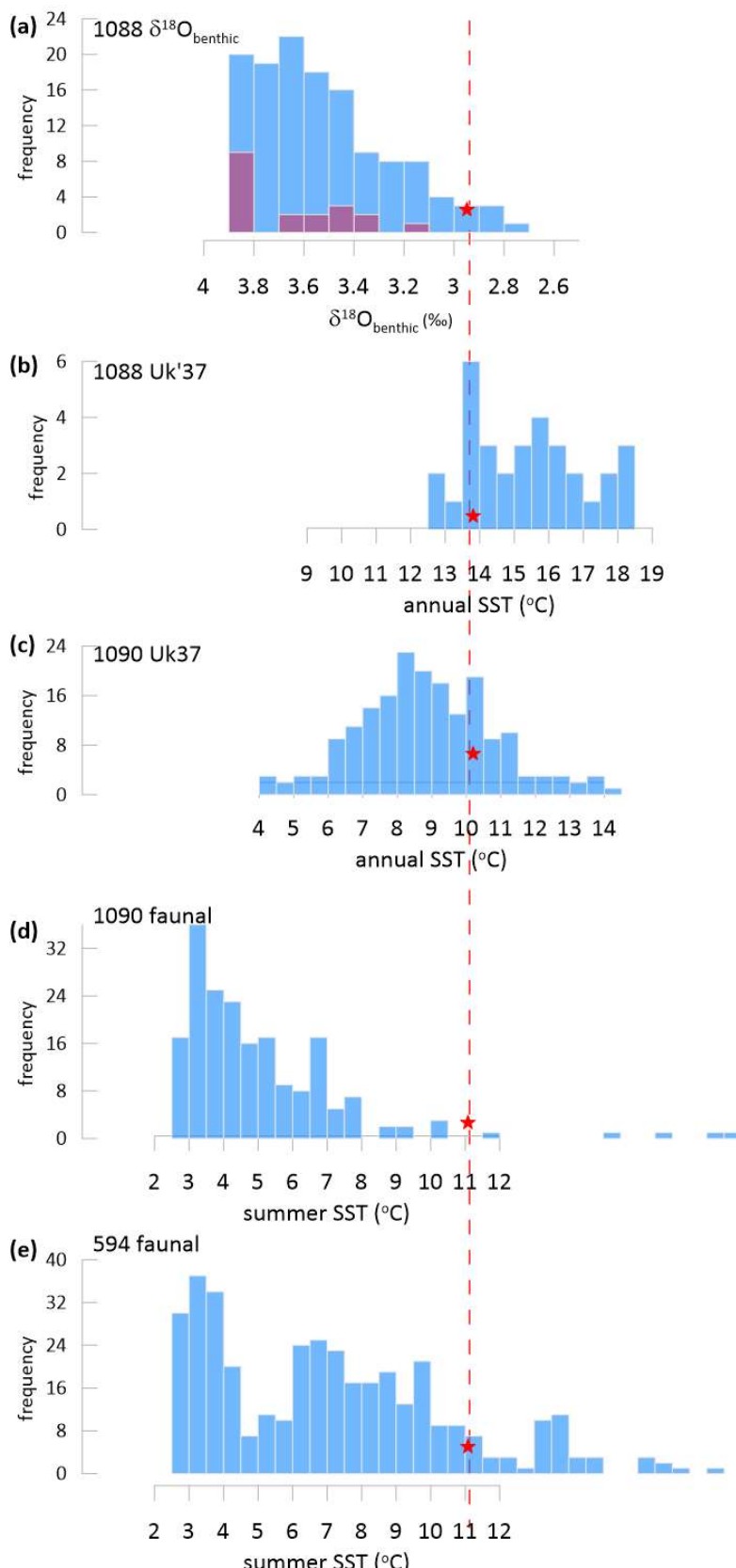

Figure 5. (a) –(e). Histograms showing the distribution of South Atlantic and Southern Ocean proxy estimates over the last 1 Ma compared to the current interglacial value (red star). (a) Histogram of benthic $\delta^{18}O$ with blue showing all data (Hodell et al., 2003) with the purple overlay showing the distribution for samples employed in Uk'37 analysis of Site 1088; Holocene benthic $\delta^{18}O$ from (Mackensen et al., 2001). (b) and (c) alkenone SST estimates from Site 1088 and 1090 (Herbert et al., 2016; Martínez-Garcia et al., 2009). (d) and (e) show summer SST estimated from foraminiferal assemblages at Site 1090 and site 594 which is located between the Subtropical and Subpolar fronts in the Indo-Pacific Ocean (Becquey and Gersonde, 2002; Schaefer et al., 2005).

### 4.1.5 Proxy temperature distribution and calibration biases

Warm biases are recognized in the calibration of core top the $U_{37}^{k\prime}$ to mean annual SST (Tierney and Tingley, 2018). The benthic $\delta^{18}O$ distribution for Site 1088 highlights that most of the samples for which $U_{37}^{k\prime}$ has been reported correspond to glacial or intermediate (but not interglacial) climate states (Fig. 5a). In contrast, in the alkenone record from Site 1088 spanning the last 1 Ma, most of the reported $U_{37}^{k\prime}$ temperatures are warmer than the current interglacial core top (Fig. 5b, Figure S2).

In contrast, the the $U_{37}^{k}$ temperatures over the last 1 Ma at ODP 1090 are predominantly colder than the current interglacial, a distribution more representative of the actual climate variation (Fig. 5c). At Site 1090, published temperatures employ the $U_{37}^{k}$ ratio which is calculated with C37:2, C37:3 and the C37:4 alkenones. Because the C37:4 becomes more abundant at cold temperatures, the $U_{37}^{k}$ index differs significantly from the $U_{37}^{k\prime}$ index at temperatures of 10°C and lower. While the significance of non-thermal effects on the C37:4 alkenone has been discussed, recent work documents that increased abundance of C37:4 in marine alkenone producers is not due to salinity sensitivity(Liao et al., 2023; Zhang et al., 2023) but because the cell's biochemical response to temperature involves adjustment of the entire suite of alkenones, not only the ratio of the di- to tri-unsaturated C37 methyl alkenones(Conte et al., 1998). Consequently, glacial coolings in the Southern Ocean are captured with the the $U_{37}^{k}$ index which accounts for C37:4 but not the the $U_{37}^{k\prime}$ index (Ho et al., 2012). Like the the $U_{37}^{k}$, the faunal estimates of summer SST at Site 1090 over the last 1 Ma are also predominantly colder than the current interglacial (Fig. 5d) (Becquey and Gersonde, 2002). In the Indo-Pacific ocean Site 594 (Fig. 5e), a similar site position between the subtropical and subpolar fronts in the Indo-Pacific ocean, faunal temperatures are also predominantly colder than the current interglacial (Schaefer et al., 2005). An aggregate estimate of the global mean surface temperature (GMST) also indicates temperatures dominantly colder than present interglacial for most of the last 5 Ma(Clark et al., 2024b). Given the close relationship between GMST and high latitude SSTs (Evans et al., 2024), this also suggests a likely distribution of colder than modern SST in the Southern Hemisphere high latitude ocean. This comparison suggests that the calculated $U_{37}^{k\prime}$ SSTs at Site 1088 may feature significant 4 to 5°C warm bias in the calibration, leading to overestimated temperatures for the last 1 and 5 Ma, whereas such a warm bias is not identified in the Site 1090 $U_{37}^{k}$ SST record. The $U_{37}^{k\prime}$ core top calibration applied at Site 1088 is similar to that for cultured *G. huxleyi* strain 55 of (Prahl et al., 1988) but culture of other strains in other environmental conditions reveals an array of SST-$U_{37}^{k\prime}$ intercepts and application of other culture relationships would yield even warmer temperature estimates (Conte et al., 1998; D'andrea et al., 2016). A comparison of faunal and alkenone SSTs in the Southern Ocean over the last 200 ka also suggested a consistent warm bias in the alkenone SST relative to foraminifera-based SSTs (Chandler and Langebroek, 2021).

### 4.1.6 Reconciling alkenone and clumped isotope temperatures

We have assessed that $\Delta_{47}$ coccolith temperature estimates at Site 1088 may be up to 2°C cold biased due to diagenesis. Comparison with temperature estimates over the last 1 Ma suggests that the calculated the $U_{37}^{k\prime}$ temperatures at Site 1088 may have a warm calibration bias of 4 to 5°C relative to mean annual temperature. Thus the 9°C temperature difference between coccolith production temperature and alkenone-estimated SST may be reduced to 2 to 3°C, which is within estimates of the modern difference between mean annual SST and the spring production temperatures in the upper 150m of the water column recorded by $\Delta_{47}$ coccolith temperatures.

In the past, it is possible that there were changes in the structure of the upper water column or shifts in the seasonality of phytoplankton spring blooms. Such changes may alter the relationship between the temperature at which both coccoliths and alkenones are produced and the temperature at the sea surface, an aspect that should be addressed in future model-data comparisons and consideration of the proxy calibration basis.

**4.2 Implications for estimation of latitudinal temperature gradients**

Unlike the higher latitude estimates for which coccolith $\Delta_{47}$ temperatures are significantly cooler than $U^{k\prime}_{37}$ SST , coccolith $\Delta_{47}$ estimates of upper ocean temperatures in the tropical Atlantic Site 926 in the Late Miocene range from 26.5 to 35.2°C (Tanner et al., 2025) (Fig. 6a). At that site in the Late Miocene, the alkenone the $U^{k\prime}_{37}$ index is saturated, indicating temperatures warmer than 28-29°C (Tanner et al., 2025). This indicates that $\Delta_{47}$ coccolith temperatures are not universally colder than alkenone temperatures, especially when accounting for the difference between SST and production temperature. For the periods in which we have overlapping data from Site 1088 and Site 926, the $U^{k\prime}_{37}$ SST implies a minimum 8°C temperature gradient between the two sites, whereas the $\Delta_{47}$ of coccoliths implies a nearly 16°C temperature gradient between the two site (Fig. 6a).

Comparison of clumped isotope temperatures on planktic foraminifera and that $U^{k\prime}_{37}$ SST elsewhere in the southern Ocean during warm climates earlier in the Miocene (12-12.5Ma) also suggest a warm bias in the $U^{k\prime}_{37}$ SST which may underestimate southern hemisphere latitudinal temperature gradients. As observed at Site 1088 in the South Atlantic, in the high latitude South Pacific Ocean, Miocene $\Delta_{47}$ upper ocean temperature estimates are significantly (14°C) cooler than estimates from the $U^{k\prime}_{37}$ SST (Fig. 6b). In the 12 to 12.5 Ma time interval in which the records overlap, $U^{k\prime}_{37}$ SST estimates from Deep Sea Drilling Project (DSDP) Site 594 (Latitude 45° 30′ S, Longitude 174° E) suggest temperatures of 24°C(Herbert et al., 2016), whereas $\Delta_{47}$ temperatures from well-preserved planktic foraminifera *Globigerina bulloides* from ODP Site 1171 (Latitude 48° 30′ S, Longitude 149° 07′ E) provide calcification temperature estimates of 11°C(Leutert et al., 2020). In this setting, foraminifera *G. bulloides*, like alkenone-producing haptophytes, has highest production in austral spring. Other temperature proxy records from Site 1171 feature significant uncertainty due to multiple calibrations and challenges estimating past seawater Mg/Ca (Fig. 6b). In the high latitude North Atlantic, $\Delta_{47}$ of coccoliths also indicates much cooler temperatures than the $U^{k\prime}_{37}$ SST(Mejia-Ramirez et al., 2024).

Latitudinal sea surface temperature gradients affect the strength of the atmospheric (Hadley cell) circulation as well as the upper ocean vertical stratification(Boccaletti et al., 2004). If there is a widespread overestimation of high latitude temperatures and underestimation of latitudinal temperature gradients during past warm periods such as the Pliocene or Miocene, this would have several implications for data model comparisons. Pacific latitudinal temperature gradients in models and proxies have been compared using high latitude temperature as an index of climate state (Liu et al., 2022), but if absolute high latitude temperatures are overestimated by proxies, then an alternate set of model characteristics (such as climate sensitivity) may provide a better match to observations. Because of the influence of latitudinal SST gradient on atmospheric circulation and precipitation patterns, some model data comparisons have imposed a proxy-based SST gradient in an effort to generate more consistent model-data comparisons (Lu et al., 2021; Burls and Fedorov, 2017), and the robustness of this imposed SST pattern would need to be reassessed. Additionally, tuning of model latitudinal gradients in cloud properties such as cloud albedo is one mechanism which has been applied to reduce the model-data discrepancy in latitudinal SST gradients (Fedorov et al., 2015), but a revision of proxy latitudinal gradients may necessitate reconsideration of the scope of feedbacks required to simulate polar amplification in past warmer climates, including not only the Miocene but also warm intervals of the Plio-Pleistocene. Finally, an overestimation of sea surface temperature may also lead to overestimation of atmospheric $CO_2$ concentrations from proxies which directly reconstruct $[CO_2]_{aq}$, such as phytoplankton isotope fractionation or boron isotopes. For example, a reduction in SST from 21°C to 16°C would reduce the estimated $p$CO$_2$ from a given $[CO_2]_{aq}$, by ≈13% due to the higher gas solubility at colder temperatures. Our analysis suggests that further assessment of absolute proxy temperature estimates and their calibrations is needed before robust model-data comparisons can be carried out.

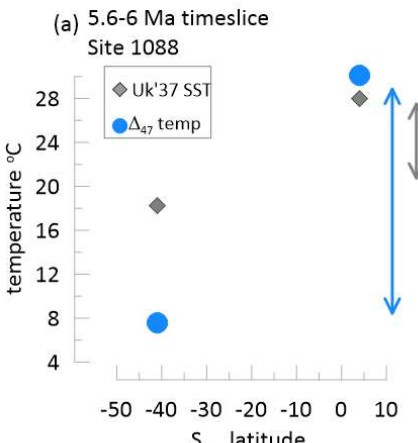

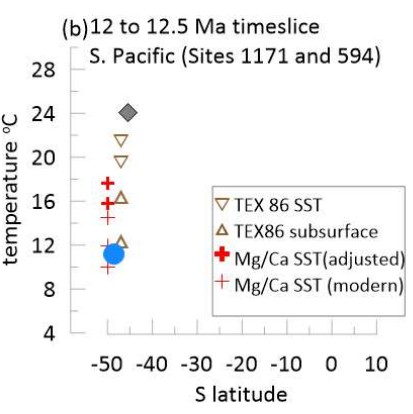

**Figure 6: Comparison of $\Delta_{47}$ upper ocean temperature estimates with alkenone temperature estimates. (a)** compares $\Delta_{47}$ coccolith temperatures averaged over the 5.6 to 6 Ma time interval from Site 1088 (this study) and low latitude Atlantic Site 926 (Tanner et al., 2025) to the $U_{37}^{k'}$ temperatures from Site 1088 (Herbert et al., 2016) and Site 926 (Tanner et al., 2025) **(b)** compares $\Delta_{47}$ temperatures from planktic foraminifera *G. bulloides* (Leutert et al., 2020) at Site 1171 to the $U_{37}^{k'}$ temperatures from nearby DSDP Site 594 (Herbert et al., 2016) for which the modern mean annual SST differs by <1°C (Levitus et al., 1994); also shown from Site 1171 are TEX₈₆ temperature estimates from Leutert et al. (2020) using calibration to SST (Tierney and Tingley, 2015; Kim et al., 2010) and calibration to subsurface temperature (Ho and Laepple, 2016; Tierney and Tingley, 2015), and Mg/Ca temperatures from *G. bulloides* (Shevenell et al., 2006; Shevenell et al., 2004) recalculated by Leutert et al. (2020) with three calibrations (Gray and Evans, 2019; Vázquez Riveiros et al., 2016; Mashiotta et al., 1999) assuming modern seawater Mg/Ca and a proposed scenario for Miocene seawater Mg/Ca(Lear et al., 2015). The TEX₈₆ and Mg/Ca temperatures from Site 1171 are plotted with a slight + and - 1.5° latitudinal offset, respectively to improve clarity in the figure.

## 5. Conclusions

New determinations of upper ocean temperatures in the South Atlantic at Site 1088 using $\Delta_{47}$ thermometry on coccoliths shows that the coccolith production temperatures declined from 14.6 °C at 15 Ma during the Miocene Climatic Optimum to 4.6°C during a glacial period in the early Pleistocene. While this magnitude of cooling is consistent with that seen in SST estimated from the $U_{37}^{k'}$ SST at Site 1088, the absolute temperatures derived from $\Delta_{47}$ thermometry on coccoliths are 9°C colder. The $\Delta_{47}$ temperatures were measured on a size fraction of coccoliths dominated by coccoliths of alkenone producers, suggesting that differences in proxy calibration and sedimentary alteration, and not differences in production season or niche, are responsible for the differences. We suggest that a 4-5°C warm bias in the alkenone calibration for this site, together with the potential for a slight 2°C cold bias in the $\Delta_{47}$ due to minor diagenetic overgrowth, could reconcile these temperature differences considering a 3°C difference between production temperature and SST at this site.. The significant deviations in reconstructed absolute temperatures pose a challenge because most climate model-data comparisons are based on comparison of absolute proxy and model temperatures. Robust simulation of atmospheric circulation patterns including rainfall distribution require accurate estimates oftemperature gradients on land and in the ocean (Burls and Fedorov, 2017), and the prediction of high latitude ice sheet stability depends on accurate estimates of high latitude temperature amplification (Gasson et al., 2013). The results of this study suggest that while proxies show high fidelity in reconstructing past temperature trends, the issue of absolute temperature estimation, crucial to evaluation of models, requires continued scrutiny

**Data availability**: New $\Delta_{47}$ coccolith temperature determinations and coccolith assemblages are summarized in Supplementary Tables 1 and 2. Full instrumental data on replicates is archived at the ETH Research Collection doi 10.3929/ethz-b-000737537.

**Supplementary information** accompanies this manuscript.

**Author contributions:** The study was conceived by HS and SB. CB completed the 1088 separations and evaluated the analyzed coccolith assemblages at 1088 and 1090. MJ prepared samples for clumped isotope analysis and completed analysis

under the direction of SB. All authors discussed interpretations. The manuscript was written by HS, with input from SB and CB.

**Competing interest:** The authors declare that they have no conflict of interest.

**Acknowledgments:** We acknowledge provision of samples from the International Ocean Drilling Program.

**Financial support:** Funding for HS and SB from Swiss National Science Foundation (Award **10000231).**

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
