# Peer review of "Divergent estimates of Miocene to Pleistocene upper ocean temperatures in the South Atlantic Ocean from alkenone and coccolith clumped isotope proxies"

_EGUsphere, 2025_

## Referee Comment (RC1)

DIVERGENT ESTIMATES OF MIOCENE TO PLEISTOCENE UPPER OCEAN TEMPERATURES IN THE SOUTH ATLANTIC OCEAN FROM ALKENONE AND COCCOLITH CLUMPED ISOTOPE PROXIES- EGUSPHERE

0. Abstract
    a. The abstract does a good job at outlining the purpose and results of the paper
1. Introduction
    a. You start by talking about how latitudinal temp gradients are important for atmospheric circulation, rainfall, etc. I'd also come back to this at the end, either in the conclusion or in section 4.2, just to bring it full circle. I have notes on this below
    b. Line 54: I'm guessing you submitted this while Clark et al. 2025 was in review, but I think it's out now, so the citation should be updated
2. Settings, Sediments, and Analytical Methods
    a. Sample preparations and analysis
        i. I'm not super familiar with clumped isotope preparation, when you say 'oxidized for 5 hour or 8-14 hour' (line 101), I'm assuming there was a reason some samples required the longer reaction time. Was this simply a matter of looking at the sample at hour 5 and noting there were still organics present, thus they were oxidized longer?
            1. Ok, after reading further in section 3 and the supplemental, it sounds like two samples from each depth underwent oxidation to compare the 5 hr oxidation time to the longer 8-14 hr time. Can you clarify this in the methods? Also, I noticed that you didn't do this for each sample, only a few, is this due to sediment availability or was there another reason you chose to do the 5 hr vs overnight oxidation comparison for these specific samples vs the others?
3. Results
    a. Line 150: "the three samples with minor detrital carbonate… temps are 1-4 degrees warmer than Uk37." This is hard to tell from figure 3d, can you add a running average for the ODP 1090 Uk37 like you did for the ODP 1088 temps? Or a table to directly show the clumped isotope temps vs uk37 temps?
    b. Unless I missed it in the text, I don't believe you discuss in depth the potential influence of detrital carbonate on your temperature calculations. Is there a connection between the detrital carbonate inclusion and warmer

temperature calculations for those three samples from ODP 1090? Are they less likely to be accurate because of the presence of carbonates?

    c. Line 209-213: I feel like there should be a citation or reference here

    d. Line 216: Not sure what the convention is in EGUsphere for referring to figures within citations, but I think this citation needs to be either ((Bolton and Stoll, 2013) Figs S6, S7) or (Bolton and Stoll, 2013; Figs S6, S7)

    e. Implications for estimation of latitudinal temperature gradients

        i. That's a huge difference in the temperature gradient, something that would certainly have impacts on both atmospheric and oceanic circulations. Considering the role that a changing latitudinal temperature gradient is thought to pay in the intensification of Northern Hemisphere Glaciation and, eventually, the Mid-Pleistocene Transition, I think it would be interesting to include a comparison of Uk37 and clumped isotope temperatures for the ~1Ma timeslice, however I know this is a new technique and the data might not be available yet. Maybe you could include a sentence or two in this section or the conclusions to say that it's not just the Miocene's latitudinal temp gradient that could be stronger, it could be other time periods as well. The implications of this are big for time periods in which a shifting latitudinal temp gradient are important (like the MPT). This might inspire further research.

        ii. I'd also like to see more of a discussion of whether such a dramatic meridional temperature gradient would impact our current understanding of late Miocene climate, considering the new data suggests the gradient is almost double what the previous data suggested.

        iii. How do your reconstructed temperatures compare to Mg/Ca ratios from the Southern Ocean or South Atlantic? Other proxies? Do these proxies more closely agree with clumped isotopes or alkenone proxies?

4. Conclusions

    a. Again, I'd circle back to your point at the beginning that latitudinal temp gradients are vital for atmospheric circulation, rainfall etc. Emphasize the importance of getting this gradient accurate for other fields

5. Figures:

    a. Fig 1: Are the contours temperature? If so, can you include that in the caption?

b.  Fig 3: As I mentioned above, would it be possible to add a  running average for the ODP 1090 Uk37  like you did for the ODP 1088 temps?

---

## Author Comment (AC1)

**Reviewer 2_Comments in black, Responses in Violet**

DIVERGENT ESTIMATES OF MIOCENE TO PLEISTOCENE UPPER OCEAN TEMPERATURES IN THE SOUTH ATLANTIC OCEAN FROM ALKENONE AND COCCOLITH CLUMPED ISOTOPE PROXIES- EGUSPHERE

0. Abstract
   a. The abstract does a good job at outlining the purpose and results of the paper

1. Introduction
   a. You start by talking about how latitudinal temp gradients are important for atmospheric circulation, rainfall, etc. I'd also come back to this at the end, either in the conclusion or in section 4.2, just to bring it full circle. I have notes on this below
   b. Line 54: I'm guessing you submitted this while Clark et al. 2025 was in review, but I think it's out now, so the citation should be updated
      The citation has been updated in the revised text.

2. Settings, Sediments, and Analytical Methods
   a. Sample preparations and analysis
      i. I'm not super familiar with clumped isotope preparation, when you say 'oxidized for 5 hour or 8-14 hour' (line 101), I'm assuming there was a reason some samples required the longer reaction time. Was this simply a matter of looking at the sample at hour 5 and noting there were still organics present, thus they were oxidized longer?
         1. Ok, after reading further in section 3 and the supplemental, it sounds like two samples from each depth underwent oxidation to compare the 5 hr oxidation time to the longer 8-14 hr time. Can you clarify this in the methods? Also, I noticed that you didn't do this for each sample, only a few, is this due to sediment availability or was there another reason you chose to do the 5 hr vs overnight oxidation comparison for these specific samples vs the others?

We clarify this by adding the following to section 2.2 :
To reduce potential interferences by organic phases during measurement, sediments were oxidized with 10% $H_2O_2$ buffered solution with $NH_3$ to pH=8-9 and the oxidant removed. Where the abundance of material in the size fraction allowed, two oxidation times were compared, a 5-hour long oxidation and an 8–14-hour long oxidation, and where material was limiting the 5-hour long oxidation was performed.

3. Results
   a. Line 150: "the three samples with minor detrital carbonate... temps are 1-4 degrees warmer than Uk37." This is hard to tell from figure 3d, can you add a running average for the ODP 1090 Uk37 like you did for the ODP 1088 temps?

Or a table to directly show the clumped isotope temps vs uk37 temps?

We thank the reviewer for this suggestion. We add a LOESS smooth also for the Site 1090 Uk37 SST record (see figure at end of this document).

b.  Unless I missed it in the text, I don't believe you discuss in depth the potential influence of detrital carbonate on your temperature calculations. Is there a connection between the detrital carbonate inclusion and warmer temperature calculations for those three samples from ODP 1090? Are they less likely to be accurate because of the presence of carbonates?

We clarify this in section 3.1 adding the following:

Wind transport of detrital minerals to this location via strong westerlies has been documented from Patagonia and southern South America as well as Southern Africa (Barkley et al., 2024). For the cold coccolith temperatures (5 °C), small amounts (<5%) of detrital carbonate would warm the measured clumped isotope temperatures by <1.5°C if the clumped isotope signature of detrital carbonates reflected temperatures of carbonate precipitation in earth surface environments at temperatures <35°C. However, if the source of the detrital carbonates has had clumped temperatures reset to much higher burial temperatures (>100°C) it could shift the measured temperatures warmer by several degrees in coccolith fractions containing >3% of such detrital carbonates. Since the origin and burial history of the detrital carbonates cannot be readily constrained for this setting, and they cannot be effectively isolated from the coccoliths for analysis of their clumped temperature, it is not possible to predict if they appreciably impact the measured temperatures in the samples containing them. As a conservative approach, we do not make further interpretations from these samples.

c.  Line 209-213: I feel like there should be a citation or reference here – we add the reference to the temperature profiles, so the revised paragraph reads:

At Site 1088, like most regions of the ocean, upper ocean temperatures are warmer than those at the seafloor (Locarnini et al., 2013). If the measured coccolith carbonate represented a mixture of primary biogenic calcite produced in the euphotic zone and secondary diagenetic carbonate precipitated in the colder waters on the seafloor, this could cause coccolith $\Delta_{47}$ temperatures to be significantly colder than those in the euphotic zone. However, in the higher latitudes such as Site 1088, because of the relatively modest temperature gradient between the seafloor and the euphotic zone (Locarnini et al., 2013), extreme degrees of diagenetic overgrowth would be required to significantly shift the temperature signal of the measured carbonate.

d.  Line 216: Not sure what the convention is in EGUsphere for referring to figures within citations, but I think this citation needs to be either ((Bolton and Stoll, 2013) Figs S6, S7) or (Bolton and Stoll, 2013; Figs S6, S7) we adjust accordingly

e.  Implications for estimation of latitudinal temperature gradients

    i.  That's a huge difference in the temperature gradient, something that would certainly have impacts on both atmospheric and oceanic circulations. Considering the role that a changing latitudinal temperature gradient is thought to pay in the intensification of Northern Hemisphere Glaciation and, eventually, the Mid-Pleistocene Transition, I think it would be interesting to include a comparison of

Uk37 and clumped isotope temperatures for the ~1Ma timeslice, however I know this is a new technique and the data might not be available yet. Maybe you could include a sentence or two in this section or the conclusions to say that it's not just the Miocene's latitudinal temp gradient that could be stronger, it could be other time periods as well. The implications of this are big for time periods in which a shifting latitudinal temp gradient are important (like the MPT). This might inspire further research.

We appreciate the suggestion from the reviewer. Indeed, there are very limited surface ocean temperature estimates available from clumped isotopes in planktic foraminifera or coccoliths at this time; in the last 1 myr the new data in this paper and in the cited Mejia et al. (preprint), so it is not possible yet to include a compilation. However, the suggestion to add a comment in the conclusions about testing latitudinal gradients at other times is one we would be happy to add. We add the bold phrase to this sentence in the end of section 4.2:

…a revision of proxy latitudinal gradients may necessitate reconsideration of the scope of feedbacks required to simulate polar amplification in past warmer climates, **including not only the Miocene but also the Plio-Pleistocene.**

ii.  I'd also like to see more of a discussion of whether such a dramatic meridional temperature gradient would impact our current understanding of late Miocene climate, considering the new data suggests the gradient is almost double what the previous data suggested.

We add the following at the end of section 4.2:

Latitudinal sea surface temperature gradients affect the strength of the atmospheric (Hadley cell) circulation as well as the upper ocean vertical stratification (Boccaletti et al., 2004). If there is a widespread overestimation of high latitude temperatures and underestimation of latitudinal temperature gradients during past warm periods such as the Pliocene or Miocene, this would have several implications for data model comparisons. Pacific latitudinal temperature gradients in models and proxies have been compared using high latitude temperature as an index of climate state (Liu et al., 2022), but if absolute high latitude temperatures are overestimated by proxies, then an alternate set of model characteristics (such as climate sensitivity) may provide a better match to observations. Because of the influence of latitudinal SST gradient on atmospheric circulation and precipitation patterns, some model data comparisons have imposed a proxy-based SST gradient in an effort to generate more consistent model-data comparisons (Lu et al., 2021; Burls and Fedorov, 2017), and the robustness of this imposed SST pattern would need to be reassessed. Additionally, tuning of model latitudinal gradients in cloud properties such as cloud albedo is one mechanism which has been applied to reduce the model-data discrepancy in latitudinal SST gradients (Fedorov et al., 2015), but a revision of proxy latitudinal gradients may necessitate reconsideration of the scope of feedbacks required to simulate polar amplification in past warmer climates, including not only the Miocene but also warm intervals of the Plio-Pleistocene. Finally, an overestimation of sea surface temperature may also lead to overestimation of atmospheric $CO_2$ concentrations from proxies which directly reconstruct $[CO_2]_{aq}$, such as phytoplankton carbon isotope fractionation or boron isotopes. For example, a reduction in SST from 21°C to 16°C would reduce the estimated $pCO_2$ from a given $[CO_2]_{aq}$, by ≈13% due to the higher gas

solubility at colder temperatures. Our analysis suggests that further assessment of absolute proxy temperature estimates and their calibrations is needed before robust model-data comparisons can be carried out.

iii. How do your reconstructed temperatures compare to Mg/Ca ratios from the Southern Ocean or South Atlantic? Other proxies? Do these proxies more closely agree with clumped isotopes or alkenone proxies?

We thank the reviewer for suggesting that we increase the comparison to other proxies. From the core in which we produced our new time series, Site 1088, existing surface ocean temperature estimates are available only from alkenones and the new clumped isotope records. From the Southern Ocean core Site 1171, in which planktic foraminiferal clumped isotope temperatures were illustrated in Figure 6b, there are additionally TEX86 determinations (Leutert et al., 2020) and Mg/Ca on the planktic foraminiferal species *G. bulloides* (Shevenell et al 2004, 2006, recalculated in Leutert et al 2020). We therefore propose to include the Site 1171 TEX86 and Mg/Ca records in Figure 6b. As noted in Leutert et al, 2020 the choice of calibration for TEX86 has a strong influence on the absolute temperatures, so we illustrate in the figure all four temperature calibrations presented in Leutert et al (2020). Additionally, for the temperatures calculated from planktic foraminiferal Mg/Ca we illustrate the six SST calculations presented in Leutert et al (2020), which include three different calibrations for *G. bulloides* and three different scenarios for Mg/Ca seawater values. Because of the calibration and Mg/Ca seawater issues affecting TEX86 and planktic Mg/Ca, the inclusion of these results does not serve to "validate" clumped vs alkenone temperatures but rather illustrate the challenges in delineating robust absolute temperatures that has motivated previous studies (e.g. Leutert et al 2020) to focus on trends rather than absolute values from the TEX86 and Mg/Ca proxies.

[Figure]

[Figure]

**Figure 6: Comparison of Δ47 upper ocean temperature estimates with alkenone temperature estimates. (a) compares Δ 47 coccolith temperatures averaged over the 5.6 to 6 Ma time interval from Site 1088 (this study) and low latitude Atlantic ODP Site 926 (Tanner et al., in prep. ) to the U_37k' temperatures from Site 1088 (Herbert et al., 2016) and Site 926 (Tanner et al., in prep. ). (b) compares Δ 47 temperatures from planktic foraminifera G. bulloides (Leutert et al., 2020) at Site 1171 with to the U_37k' temperatures from nearby Site 594 (Herbert et al., 2016) for which the modern mean annual SST differs by <1°C (Levitus et al., 1994); also shown from Site 1171 are TEX86 temperature estimates from Leutert et al. (2020) using calibration to SST (Tierney and Tingley, 2015; Kim et al., 2010) and calibration to subsurface temperature (Ho and Laepple, 2016; Tierney and Tingley, 2015), and Mg/Ca temperatures from *G. bulloides* (Shevenell et al., 2006; Shevenell et al., 2004) recalculated by Leutert et al. (2020) with three calibrations (Gray and Evans, 2019; Vázquez Riveiros et al., 2016; Mashiotta et al., 1999) assuming modern seawater Mg/Ca and a proposed scenario for Miocene seawater Mg/Ca (Lear et al., 2015). The TEX86 and Mg/Ca temperatures from Site 1171 are plotted with a + and -1.5° latitudinal offset, respectively, to improve clarity in the figure.**

4. Conclusions

   a. Again, I'd circle back to your point at the beginning that latitudinal temp gradients are vital for atmospheric circulation, rainfall etc. Emphasize the importance of getting this gradient accurate for other fields

   We expand the end of the conclusion to the following:

   The significant deviations in reconstructed absolute temperatures pose a challenge because most climate model-data comparisons are based on comparison of absolute proxy and model temperatures. Robust simulation of atmospheric circulation patterns including rainfall distribution require accurate estimates of temperature gradients on land and in the ocean (Burls and Fedorov, 2017), and the prediction of high latitude ice sheet stability depends on accurate estimates of high latitude temperature amplification (Gasson et al., 2013). The results of this study suggest that while proxies show high fidelity in reconstructing past temperature trends, the issue of absolute temperature estimation, crucial to evaluation of models, requires continued scrutiny.

5. Figures:

   a. Fig 1: Are the contours temperature? If so, can you include that in the caption?

   Good point, the revised figure caption notes this.

b.  Fig 3: As I mentioned above, would it be possible to add a running average for the ODP 1090 Uk37 like you did for the ODP 1088 temps? – we add the LOESS fit and the figure now appears like this:

---

## Author Comment (AC2)

**Reviewer 2_Comments in black, Responses in Violet**

Stoll et al. present an interesting multi-proxy comparison of upper ocean temperatures over the last 15 million years, combining new coccolith clumped isotope-based temperature estimates with published alkenone-based estimates from the same sites. The fact that both proxies are produced by the same organism, but otherwise based on very different principles, makes this comparison especially interesting because it rules out several possible reasons for discrepancies. Yet, the paper presents a huge difference in temperature estimates from the two proxies. Several lines of reasoning including comparison with other evidence suggest that the colder clumped isotope based estimates are more realistic (though possibly slightly cold biased), whereas the Uk'37 based estimates probably suffer from a substantial warm bias. This result has important implications for the interpretation of previous alkenone-based temperature estimates from high latitudes and thus for previous estimates of latitudinal temperature gradients.

The data and message presented here are very clear and I have only minor suggestions for further improvements.

Section 4.1.5 about a suggested warm bias of the Uk'37 data from Site 1088 could be made clearer. If I understand correctly, the bias is suggested to only be present at Site 1088, not 1090, because at the latter site a different index is used (Uk37). This interpretation could be made clearer in this section and it would help non-alkenone-experts like me if the difference between different alkenone indices was introduced earlier in the paper (e.g. before the data are presented in Figure 3).

I was also left wondering whether it would be possible to recalculate the respective other index out of raw alkenone data from either site.

Another question I have is whether the potential calibration bias due to calibration versus SST rather than production depth temperature (mentioned in section 4.1.1) is specific for either of the two indices or applies to both? It would also be good to know whether using different alkenone calibrations (e.g., culture-based) change the picture in any significant way? In the same vein, it could be added that on the clumped isotope side, any other calibration choice would make the D47 temperatures even colder.

We thank the reviewer for prompting us to clarify the different alkenone indices and their application here. We add the following at the end of section 2.2:

Alkenone temperatures for Site 1088 are calculated with the Uk'37 index based on the C37:2 and C37:3 alkenone abundances; no C37:4 abundances are reported for these samples. This record employed the calibration based on regression of sea surface temperatures to globally distributed core top Uk'37 (Müller et al., 1998). Alkenone temperatures for Site 1090 employ the Uk37 index which is based on the C37:2, C37:3, and C37:4 abundances. This latter index is proposed to be better suited for colder temperature settings (Ho et al., 2012) as long as contributions from non-marine haptophytes can be ruled out (Kaiser et al., 2019). The temperature calculation at Site 1090 employs a calibration based on cultures of a strain of *Gephyrocapsa (Emiliania) huxleyi*

(Prahl et al., 1988) which is not significantly different from the calibration obtained from regressions between SST and Uk'37 in core top sediments (Sikes et al., 1991; Ho et al., 2012).

And the following lines at the end of section 4.1.5

The Uk37 core-top calibration applied at Site 1088 is similar to that for cultured *G. huxleyi* strain 55 of (Prahl et al., 1988) but cultures of other strains in other environmental conditions reveals an array of SST-Uk37 intercepts and application of other culture relationships would yield even warmer temperature estimates (Conte et al., 1998; D'andrea et al., 2016).

Finally in the end of section 4.1.5, adding the phrase in bold below:

This comparison suggests that the calculated the Uk37' SSTs at Site 1088 may feature a significant warm bias in the calibration, leading to overestimated temperatures for the last 1 and 5 Ma, **whereas such a warm bias is not identified in the Site 1090 Uk37 SST record.**

Section 4.2: The comparison with the proxy difference at Sites 1171 and 594 seems to be better placed earlier as additional argument for warm bias in the Uk'37 estimates, rather than in this section about latitudinal gradients. That would allow this section to focus on the implication for reconstructed latitudinal gradients, introduced in the introduction as a major motivation for this work. The figure (Fig 6b) could instead be incorporated into Figure 3.

We appreciate the suggestion to move figure 6b to Figure 3. At the same time, its inclusion in section 4.2 provides an outlook that the discrepancy in estimating temperature gradients may not be limited only to Site 1088 in late Miocene time window but also apply to older and even warmer periods. Additionally, reviewer 1 suggests the inclusion of a larger set of proxies for SST which we propose to add to Figure 6. We thus propose to clarify that this part of section 4.2 addresses the question – is the alkenone warm bias seen in Site 1088 potentially characterizing other high latitude locations and time intervals?

**Minor suggestions and typos**

Line 22: «must» – maybe change to «should", if it could in principle be possible that calcification occurs at somewhat different water depth than alkenone synthesis.

We are aware of no evidence that calcification and alkenone synthesis occur at different water depths (as indicated in our response to and clarification of comment line 192 below). Given that coccolithophores do not undertake vertical migrations (like some planktonic foraminifera), we prefer to retain the original wording.

Line 45: Check TEX86 spelling - revised

Line 46: Check reference formatting revised

Line 51: Add a comma after "thermometry" revised

Line 92-93: Check reference formatting revised

Line 95: analyzed - revised

Line 101-102: Check sentence structure – this section has been rewritten in response to Reviewer 2's request to clarify the meaning, so should have clear sentence structure now

Line 111: Check reference formatting revised

Line 114: Is it correct that each sample replicate was corrected with the closest 12 standard measurements (ETH 1-3), i.e., less that a run worth of standards?

This is clarified with the following:

Each analytical run consists of 10 aliquots of ETH-3, 5 aliquots each of ETH-1 and ETH-2 organized in three blocks: one at the beginning, one in the middle and one at the end of the autosampler carousel, two aliquots of IAEA-C2 and not more than 3 aliquots each of unknown samples for a total of 22 aliquots. For each sample from Site 1088 a minimum of 14 replicates and for 1090 a minimum of 10 replicates distributed in 4 to 5 analytical runs was measured. For standardization, a moving window of 12 standards measured before and 12 standards measured after each sample (spanning two to three analytical runs) was used.

Line 116: What is meant by "batches"? – this is now replaced with the term "analytical runs"

Line 124: Give references for the alkenone records - revised

Line 141-142: How are the samples containing detrital carbonate treated for interpretation? Are they disregarded (as the crosses in Fig 2 suggest)?

In section 3.1 we now clarify:

Since the origin and burial history of the detrital carbonates cannot be readily constrained for this setting, and they cannot be effectively isolated from the coccoliths for analysis of their clumped temperature, it is not possible to predict if they appreciably impact the measured temperatures in the samples containing them. As a conservative approach, we do not make further interpretations from these samples.

Line 148: At ODP Site 1090,...revised

Line 191-192: Can it be ruled out that coccolith and alkenone production occur at different water depths? Is this statement linked to the second sentence of this section, possibly implying continuous production of both phases during the lifetime of the organisms? If so, that could be made clearer.

We have clarified the statement to confirm that both organic and inorganic phases are produced continuously during the lifetime of the organisms.

Line 193-195: Specify that the calcification temperature is known for the D47 calibration as it is based on cultures.

We agree to clarify by stating: *The calibration of $\Delta_{47}$ to temperature is made using calcification temperature based on experimentally grown cultures, and therefore reflects the temperature at which coccolithophorids grow and calcify in the photic zone, in this case during the bloom season.*

Line 196: Add space after "depth" - revised

Line 197: Add space after "temperatures" - revised

Line 199-202: relating the alkenone index to SST or even summer SST even though the signal is produced deeper in the water column or in a different season must assume that vertical or temporal differences are similar at all sites used in the calibration and where the calibration is applied, which seems problematic and could be made even more clear here. Even though the gradient is weak at Site 1088, it could be stronger in most core top locations from the calibration, which would still bias the signal too warm.

We emphasize this important point by adding: *This calibration approach assumes that vertical or temporal differences are similar at all sites used in the calibration and where the calibration is applied.*

Line 204-207: Is this only true/known for part of the record?.

We clarify,

*Therefore, at this setting in modern oceanographic conditions, the distinct temperature calibration targets may explain up to 3°C difference*

Line 216: Check reference formatting - revised

Line 216-217: Could there be other particles of similar size as the coccoliths that are produced at colder temperatures/at the seafloor?

We propose to add to the beginning of section 2.2 (which describes the composition of the size fraction) the clarification that:

*This analysis verifies that there is no significant non-coccolith biogenic or abiogenic carbonate in this size fraction.*

Line 217: Add "in our record" or "here" to make it clear that the results from this study are referred to. revised

Line 231: Add space after "biomineralization" revised

Line 266: Diagenetic processes? Instead of "cool … signals" I suggest reformulating to "introduce a cold bias to…" (same for the next part of the sentence) revised

Line 268: Check formatting of Uk37 revised

Line 271: Add space after "index" revised

Line 288-289: Check sentence structure revised

Line 311: Check sentence structure revised

Line 314: Check sentence structure revised

Line 316: Check sentence structure. Here and subsequent sentences: Check use of $U^k_{37}$ versus $U^{k'}_{37}$. revised

Line 320: The reference to a potential salinity effect comes a bit out of the blue and makes the sentence convoluted. Rather split up the sentences and introduce this potential effect better. – Revised to:

While the significance of non-thermal effects on the C37:4 alkenone has been discussed, recent work documents that increased abundance of C37:4 in marine alkenone producers is not due to salinity sensitivity(Liao et al., 2023; Zhang et al., 2023) but because the cell's biochemical response to temperature involves adjustment of the entire suite of alkenones, not only the ratio of the di- to tri-unsaturated C37 methyl alkenones(Conte et al., 1998).

Line 326: Check sentence structure with double brackets - revised

Line 335-336: the scale of suggested warm bias would be better established in the previous section where the basis for this statement is discussed, and then repeated here. – now added in section 4.1.5 as well as in 4.1.6

Line 347: Check sentence structure – the alkenone index saturation is correctly described.

Line 350: If the Uk'37 index is saturated at Site 926, the calculated gradient should be a minimum estimate - revised

Figure 2:

The figure caption could give some more detail on the different kinds of data shown in the figure to allow the reader a quicker overview. E.g., that the species listed here are dominating the respective size fractions (with reference to the supplement), that crossed out samples contain abundant detrital carbonate,... clarified

Caption for Fig 2: Panel b) is for Site 1090, not 1088. revised

Format axes titles consistently for all Figures (e.g. "Age (Ma)", "D47 temp. (°C)") revised

Figure 4: Figure caption: Check sentence for panel a). revised